# Position: The Right to AI

**Rashid Mushkani** [1 2]  **Hugo Berard** [1]  **Allison Cohen** [2]  **Shin Koseki** [1 2]

## Abstract

**This position paper proposes a "Right to AI," which asserts that individuals and communities should meaningfully participate in the development and governance of the AI systems that shape their lives.** Motivated by the increasing deployment of AI in critical domains and inspired by Henri Lefebvre's concept of the "Right to the City," we reconceptualize AI as a societal infrastructure, rather than merely a product of expert design. In this paper, we critically evaluate how generative agents, large-scale data extraction, and diverse cultural values bring new complexities to AI oversight. The paper proposes that grassroots participatory methodologies can mitigate biased outcomes and enhance social responsiveness. It asserts that data is socially produced and should be managed and owned collectively. Drawing on Sherry Arnstein's Ladder of Citizen Participation and analyzing nine case studies, the paper develops a four-tier model for the Right to AI that situates the current paradigm and envisions an aspirational future. It proposes recommendations for inclusive data ownership, transparent design processes, and stakeholder-driven oversight. We also discuss market-led and state-centric alternatives and argue that participatory approaches offer a better balance between technical efficiency and democratic legitimacy.

## 1. Introduction

**We posit that every individual and community affected by artificial intelligence (AI) systems has a *Right to AI*: the capacity and entitlement to shape, critique, and govern the AI infrastructures that increasingly define modern life.** AI is proliferating in domains such as healthcare, education, finance, and urban planning, generating both innovation and ethical, legal, and socio-political concerns (Lepri et al., 2018; Avellan et al., 2020; Larsson, 2020; Taei-hagh, 2021; de Hond et al., 2022; Queerinai et al., 2023; Huang et al., 2024a; Zhou et al., 2024; Zhang et al., 2025; Goodman & Dai, 2025). While the transformative potential of AI is evident, disparities in its design and deployment reveal patterns of algorithmic bias, challenges with algorithmic fairness, as well as risks to privacy, among other human rights concerns (Brayne, 2017; Arslan, 2017; Costanza-Chock, 2020; Shepardson et al., 2024; Cohen & Suzor, 2024; Ulnicane, 2024). Many development practices continue to prioritize efficiency and scalability at the expense of inclusion, often excluding the public from meaningful participation in AI governance (Kalluri, 2020; Sloane et al., 2022; Bengio et al., 2024; Kirk et al., 2024). The growing concentration of design decisions within a limited set of corporate and governmental entities—whether in setting priorities, allocating resources, or determining deployment practices—risks marginalizing public agency and reducing individuals to passive recipients of technological systems that increasingly shape their opportunities, well-being, and autonomy (Raz, 1999; Reisman et al., 2018; Huang et al., 2024a; Cohen & Suzor, 2024; OpenAI & SoftBank, 2025; Goodman & Dai, 2025).

We adopt Henri Lefebvre's *Right to the City* framework (Lefebvre, 1968)—which challenges top-down urban planning by emphasizing resident participation in creating livable urban spaces—and extend its spirit to the digital sphere. In this view, AI functions as *societal infrastructure*, necessitating broad-based, co-creative involvement analogous to city building or community-led educational reforms (Jacobs, 1961; Ng, 2017; Sloane et al., 2022; Birhane et al., 2022). The "ladder of citizen participation" (Arnstein, 1969) highlights how engagement can range from tokenistic consultation to meaningful empowerment, underscoring the need to address entrenched power asymmetries (Costanza-Chock, 2020; Birhane et al., 2022). As historical movements have shown, genuine participation can only emerge when communities *and stakeholders*, much like Jane Jacobs' grassroots advocacy for urban neighborhoods (Jacobs, 1961), actively organize and demand a seat at the table.

The Right to AI builds on precedents in human rights and technology. Article 27 of the *Universal Declaration of Human Rights* affirms the right to "share in scientific ad-

---

[1]Université de Montréal [2]Mila – Quebec AI Institute. Correspondence to: Rashid Mushkani <rashid.ahmad.mushkani@umontreal.ca>.

*Proceedings of the 42$^{nd}$ International Conference on Machine Learning*, Vancouver, Canada. PMLR 267, 2025. Copyright 2025 by the author(s).

vancement and its benefits," and the United Nations has recognized internet access as a fundamental right (Sun, 2020; Wenar, 2023). The Right to AI not only builds on these foundations but also *emphasizes collective empowerment* (Sun, 2020), extending the focus from mere access to a *power right* that allows individuals and communities to meaningfully influence AI systems (Wenar, 2023). When AI is trained on public data—collected from social platforms, government databases, or shared cultural artifacts—transparency and accountability become paramount to avoid a new form of "data enclosure," in which public knowledge is commercialized without returning benefits to the communities that produced it (Beer, 2016; Kitchin, 2016; Nucera & Onuoha, 2018; Lewis et al., 2020; Gerdes, 2022).

By conceptualizing AI as a collective resource, the Right to AI foregrounds public involvement in setting objectives, establishing constraints, and determining acceptable risks. This includes interrogating how personal information is collected and shared (Marmor, 2015; Wachter & Mittelstadt, 2019), ensuring that AI systems do not perpetuate statistical discrimination or erode autonomy (Kalluri, 2020; Sloane et al., 2022), and fostering mechanisms that enable broader oversight and scrutiny. In practice, it calls for a governance framework encompassing local AI councils, public audits, cooperatively managed data infrastructures, and other participatory structures that reconcile intellectual property rights with communal stewardship of AI (Ostrom, 1996; Jacobs, 1961; Lewis et al., 2020; Sun, 2020; Wenar, 2023).

**The paper's core thesis is that the optimal approach to AI governance is through a citizen-engaged process that guarantees the right to contribute, supported by a four-pronged argument for the Right to AI.** We argue that inclusive and pluralistic structures can better address biases, reflect diverse values, and strengthen democratic ideals. Section 2 situates our Right to AI proposal in the broader literature on participatory methods. Section 3 examines AI as societal infrastructure. Section 4 presents key democratic, social justice, and epistemic justifications. Section 5 introduces a four-tier model of citizen involvement, and Section 6 distills lessons from relevant participatory projects. Finally, Section 7 offers steps toward realizing the Right to AI, Section 8 addresses critiques of existing governance models, and Section 9 reflects on AI as a co-created resource.

We use the term *"Right to AI"* to emphasize the collective governance dimension of AI oversight. This governance-oriented right is broader than more familiar claims such as the right to be forgotten, the right to explanation, or the right to contest AI decisions (Kaminski, 2019; Kaminski & Urban, 2021; Zhang et al., 2024). These latter rights, while important, speak mostly to individual entitlements

to correct or clarify AI outputs. By contrast, the Right to AI we propose extends beyond mitigating harms to actively co-shaping the objectives, data practices, risk thresholds, and ethical principles of AI infrastructures. In this sense, it is more accurately viewed as a "power right" (Wenar, 2023) to guide AI's development, rather than a narrower right to information or redress.

For a deeper exploration of additional arguments—including the *Hidden Choices* analogy that compares AI to a community "kitchen," highlighting ownership, access, and accountability—see Appendices A to F, where we further discuss the broader ethical and socio-political implications of the Right to AI.

## 2. Background

### 2.1. Positioning the Right to AI

The contemporary discourse on AI governance spans policy proposals, ethical guidelines, and technical methods aimed at aligning AI with societal values (Mishra, 2023; Zaidan & Ibrahim, 2024; Sorensen et al., 2024). Institutions such as the OECD and the European Union have introduced frameworks for responsible AI development, often emphasizing fairness, accountability, and transparency (Jobin et al., 2019; Bang et al., 2024; Saheb & Saheb, 2024; Zhang et al., 2025). However, these proposals typically operate within top-down or expert-led paradigms, granting only peripheral or transactional roles to civic engagement (Jobin et al., 2019; Saheb & Saheb, 2024; Kirk et al., 2024; Huang et al., 2024a).

Recent work in *participatory AI* seeks to bridge this gap by integrating stakeholder perspectives throughout the AI lifecycle, from data collection to deployment and auditing (Sloane et al., 2022; Birhane et al., 2022; Sieber et al., 2024a). Some researchers explicitly call for *pluralistic alignment*—the notion that AI systems should be responsive to multiple moral and cultural perspectives (Sorensen et al., 2024). Yet, practical implementations often face logistical and conceptual hurdles, including defining fair representation across heterogeneous communities and reconciling conflicting values within a single system (Hoffmann et al., 2022; Mishra, 2023; Sorensen et al., 2024; Zhou et al., 2024; Kirk et al., 2024; Jin et al., 2024; Zhang et al., 2025).

### 2.2. The Right to the City

Henri Lefebvre's *Right to the City* is a foundational concept in urban theory that rejects the fragmentation of city life into discrete, expert-managed sectors. Instead, it asserts a universal right for citizens to actively shape urban processes (Lefebvre, 1968). The concept emphasizes inclusivity, accessibility, and democracy, advocating that urban spaces should be collectively governed rather than controlled solely by market forces such as commodification and capitalism.

Lefebvre's vision presents this right not as an individual entitlement but as a collective one, grounded in shared power and responsibility for shaping urban life.

Recent scholarship has expanded this framework by addressing contemporary urban struggles, emphasizing digital infrastructure, environmental justice, and participatory governance (Harvey, 2012; Purcell, 2014; Madden & Marcuse, 2017). Scholars critique the ways in which smart city initiatives, surveillance capitalism, and privatization constrain democratic urban participation. The parallels to AI governance become evident as we acknowledge AI's pervasive impact on daily life, from news curation to resource allocation (Leike et al., 2018; Koseki et al., 2022; Kitchin, 2023; Hajkowicz et al., 2023). Just as Lefebvre opposed the technocratic vision of cities as objects of specialist knowledge, the *Right to AI* challenges the notion of AI as an exclusively expert-driven endeavor.

### 2.3. Ladder of Citizen Participation

Sherry Arnstein's *ladder of citizen participation* defines eight distinct levels of citizen involvement, spanning from manipulation at the bottom to citizen power at the top (Arnstein, 1969). At the lowest rungs—manipulation and therapy—efforts aim merely to educate or "cure" participants without granting real influence. Progressing upward, forms of tokenism such as informing, consultation, and placation may appear to involve citizens, but often conceal deeper imbalances in decision-making power. The ladder serves as a guide to understanding how genuine power sharing can be distinguished from superficial involvement in decision-making processes (see Figure 1). Applied to AI, this hierarchy helps conceptualize the degree of public involvement in system design and oversight. Current findings suggest that traditional AI practices often situate users at the "informing" or "consultation" rungs at best, rarely reaching the top rungs of "partnership" or "citizen control" (Sieber et al., 2024a). Building on this framework, recent studies highlight the growing importance of civic participation and public engagement in AI (Sieber et al., 2024b). Thus, Arnstein's framework serves as a valuable lens for assessing how much decision-making power stakeholders genuinely exercise.

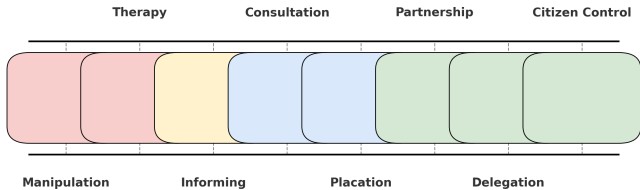

*Figure 1.* The Ladder of Citizen Participation, illustrating levels of public involvement from manipulation to citizen control.

### 2.4. Grassroots Engagement

Jane Jacobs' *The Death and Life of Great American Cities* (Jacobs, 1961) critiqued large-scale, expert-led urban redevelopment projects. Jacobs argued that community-level knowledge is often disregarded in top-down models, leading to detrimental effects on neighborhoods. Her grassroots approach resonates with the Right to AI: communities affected by AI systems also possess contextual insights that can inform more ethical, value-aligned, context-sensitive development and deployment (Arslan, 2017; Angwin et al., 2022; Birhane et al., 2022; Mushkani et al., 2025a).

Previous attempts to incorporate participation in AI governance include user feedback loops in recommender systems, collaborative training data annotation, and community reviews of AI outputs (Gerdes, 2022; Zaidan & Ibrahim, 2024; Huang et al., 2024a). Innovative proposals like "jury-based" or "constitutional" approaches also engage diverse groups in AI ethics and policy deliberations (Gordon et al., 2022; Bai et al., 2022b; Sorensen et al., 2024). However, these methods are nascent and face scalability, political, market, resource, and institutional challenges (Saheb & Saheb, 2024; Zaidan & Ibrahim, 2024; Huang et al., 2024a).

**Yet participation alone is not inherently empowering.** It can become tokenistic—what some call "participation-washing"—when stakeholders are invited without real decision-making power or follow-through. In high-stakes contexts, such as healthcare or criminal justice, participation may also be constrained by the need for technical oversight or legal accountability. These realities suggest that participatory frameworks must be adapted to the specific risks, knowledge demands, and institutional capacities of each domain. Genuine inclusion requires not just the presence of diverse voices, but mechanisms that translate deliberation into influence.

Overall, the Right to AI builds on these developments but asserts a more foundational principle: **that AI governance should not merely consult communities but *empower* them to define AI's priorities, constraints, and uses.** This shift toward recognizing AI as shared societal infrastructure underpins the arguments we develop in subsequent sections.

## 3. Arguments for AI as Societal Infrastructure

A central premise of the Right to AI is that AI increasingly functions as *societal infrastructure*, comparable to utilities or educational systems. Viewing AI as *societal infrastructure* aligns with established frameworks of public goods, commons governance, and socio-technical systems (Ostrom, 1990; Graham & Marvin, 2001). Infrastructure commonly exhibits three properties: (i) *broad societal impact*, (ii) an *essential role* in daily life, and (iii) a requirement for *collective management* (North, 1990; Davern et al., 2017). AI,

particularly foundation models shaping decisions having to do with employment, credit scoring, and public discourse, for example, arguably meets these criteria.

**Broad Societal Impact**   AI systems are being embedded as a matter of standard practice across industries such as healthcare and education, influencing areas of high social impact such as diagnostic processes and learning environments (Bommasani et al., 2022; Goodman & Dai, 2025). This pervasive integration of AI highlights the need for inclusive governance mechanisms that address the wide-ranging social implications, including ethical considerations, societal norms, and the long-term effects on communities and institutions.

**Essential Role in Daily Life**   Technologies that mediate access to financial systems, public services, and employment increasingly function as core societal infrastructure (North, 1990). AI-driven decision-making in credit approval, job screening, and social welfare administration underscores its role in structuring life chances (Eubanks, 2018; Benjamin, 2019). The opacity of these systems necessitates governance mechanisms akin to those regulating financial and legal infrastructures (Ulnicane, 2024).

**Collective Management**   AI-based systems shape social interactions, political communication, and institutional trust (Gillespie, 2018; Kitchin, 2023). Like other infrastructures, AI is not neutral; it embeds political, economic, and cultural assumptions that shape its societal consequences (Crawford, 2021; Angwin et al., 2022). Without participatory oversight, AI risks reinforcing inequities rather than serving as a mechanism for collective well-being.

Urban planning frameworks, such as the *Right to the City*, provide insights into collective governance of infrastructure, but AI differs in its algorithmic opacity and dynamic evolution (Birhane et al., 2022). Effective governance may thus require adaptive regulatory structures, participatory audits, and interdisciplinary expertise to navigate its societal impacts (Ostrom, 2009; Murray & Frijters, 2017).

## 4. Arguments for the Right to AI

The Right to AI is grounded in four distinct but overlapping arguments: *democratic legitimacy*, *social justice*, *epistemic autonomy*, and *data production*, emphasizing the necessity of community participation for ethical and effective AI.

### 4.1. Democratic Legitimacy

Democratic theories posit that decisions affecting the public should include input from those impacted (Dahl, 1971; Habermas, 1996). AI systems exert significant influence, shaping access to loans, recommending political content,

and determining university admissions. The widespread adoption of generative agents in the coming years is expected to further amplify this impact (Dastin, 2022; Jin & Zhang, 2025). To align with democratic principles, citizens must have the right to deliberate on data usage, algorithmic objectives, and mechanisms for redress (Mill, 1863; Habermas, 1996; Buruk et al., 2020). Without such participation, AI governance risks becoming an unaccountable domain controlled by elites (Benjamin, 2019).

### 4.2. Social Justice and Pluralism

Machine learning models typically generalize from large datasets, which may fail to capture minority values or nuanced cultural norms (Goodfellow et al., 2016; Gebru et al., 2021; Dastin, 2022). As a result, marginalized voices risk erasure or misrepresentation (Raz, 1999; Beer, 2016; Bondi et al., 2021; Zhang et al., 2025). The Right to AI entails inclusive governance structures that protect pluralism by ensuring that multiple moral and cultural frameworks inform system design (Lefebvre, 1968; Costanza-Chock, 2020). This pluralistic perspective challenges any hegemonic assumption that there is a single "correct" data-driven solution (Fraser, 1995; Kitchin, 2023; Sorensen et al., 2024).

### 4.3. Epistemic Autonomy

As AI systems filter information, recommend decisions, and shape daily interactions, they hold substantial power to influence knowledge ecosystems (Kalluri, 2020; Ooi et al., 2023; Huang et al., 2024a; Jin & Zhang, 2025). Epistemic autonomy refers to the ability to develop independent perspectives on what is true or valuable (Foucault, 1975; Turri et al., 2021). If AI systems are centralized or controlled by a few entities, they may homogenize culture, intensify specific worldviews, or narrow the range of acceptable discourse (Mill, 1863; Foucault, 1975; Fraser, 1995; Metz & Grant, 2024; Murgia, 2024). The Right to AI protects the capacity of individuals and communities to determine their own epistemic conditions, thereby preserving cultural diversity and safeguarding the evolution of collective knowledge (Dewey, 1927; Habermas, 1996).

### 4.4. The Production of Data

Data is integral to AI's predictive and generative capabilities (Goodfellow et al., 2016). It is created in diverse social contexts, yet the processes of collection and ownership often remain opaque and concentrated in a few organizations (Kalluri, 2020; Kitchin, 2023). These mechanisms can obscure communal contributions to datasets, allowing organizations to exercise disproportionate influence over data use. Viewing data as a shared resource aligns with Ostrom's notion of collective governance for common-pool resources (Ostrom, 1996). Approaches such as local data

trusts or transparent curation boards may mitigate risks of biased outcomes and privacy infringements by balancing innovation with individual and collective rights (Kukutai & Taylor, 2016; Lewis et al., 2020).

### 4.5. Broader Ethical Implications

Philosophical frameworks such as Design Justice (Costanza-Chock, 2020) call for marginalized community members to be at the center of the AI design and build process. The Right to AI builds on these traditions by advocating for participatory structures at each stage of the AI lifecycle (Jacobs, 1961). By distributing decision-making power and emphasizing co-ownership of data, the Right to AI embeds ethical commitments in technical artifacts and institutional arrangements (Lefebvre, 1968). Through these mechanisms, AI can better align with the values and needs of diverse communities, reinforcing social trust and ensuring that AI remains a form of shared societal infrastructure rather than a purely commercial or technocratic domain.

Moreover, at the international level, disparities in AI development create technological asymmetries between countries, shaping economic and strategic dynamics (Eubanks, 2018; OpenAI & SoftBank, 2025). Expanding access to AI models trained on publicly available data may reduce these imbalances and promote a more competitive and diverse technological landscape (Arslan, 2017).

## 5. Ladder of Right to AI

We adapt Arnstein's ladder of participation to propose four tiers of engagement in the AI governance process. These tiers are distinguished based on the *extent of stakeholder agency*, *transparency of decision-making*, and *inclusivity* in shaping AI systems (Lefebvre, 1968; Arnstein, 1969). Although these categories are not exhaustive, they illustrate a spectrum of approaches:

### 5.1. Consumer-Based (Minimal Right to AI)

In this lower tier, individuals primarily act as consumers, accessing AI services without substantive input into data practices or decision-making (Baudrillard, 1970; Kitchin, 2023). Participation is typically limited to optional user surveys or feedback forms (Kirk et al., 2024). This model offers convenience but often consolidates authority among system developers. Users have limited capacity to influence model outcomes or address biases, and redress mechanisms are generally weak (Dahl, 1971).

### 5.2. Private Organization-Led

In this tier, private entities integrate limited user feedback into governance structures that they own or manage (Dewey, 1927; Dahl, 1971; Bang et al., 2024; Zhang et al., 2025). Model behavior, training data selection, and interpretability measures remain largely within corporate purview. Transparency mechanisms (e.g., user dashboards) may partially improve accountability, but conflicts of interest can persist (Zhou et al., 2022). Communities retain a delegated form of influence, depending on the extent to which private actors incorporate public input into product roadmaps and ethical guidelines (Baudrillard, 1970; Anthropic, 2023).

### 5.3. Government-Controlled

Government agencies play a central regulatory role, setting broad guidelines that can include data privacy mandates, anti-discrimination policies, and public consultations (Habermas, 1996; Morison, 2020). This model can increase accountability by establishing enforceable standards, but top-down governance structures may overlook localized knowledge or community-specific concerns (Fischer, 2000). Moreover, government priorities may be shaped by agendas unrelated to broader stakeholder engagement, which can limit the scope of genuine participation (Jacobs, 1961; Moulin, 2004; Kukutai & Taylor, 2016; Morison, 2020).

### 5.4. Citizen-Controlled (Maximal Right to AI)

At the upper end, citizens have considerable authority over AI governance (Arnstein, 1969; Sloane et al., 2022). This model may involve local data trusts, cooperative ownership of training datasets, and citizen assemblies overseeing deployment and audit processes (Birhane et al., 2022). While such arrangements demand robust institutional support, conflict-resolution mechanisms, and technical expertise, they maximize community control (Lewis et al., 2020; Nekoto et al., 2020; Bondi et al., 2021; Birhane et al., 2022; Sloane et al., 2022). In principle, this tier represents the fullest expression of participatory AI, empowering communities to define model objectives, ethical constraints, and performance metrics (Jacobs, 1961; Arnstein, 1969).

Citizen-controlled governance envisions significant communal authority over AI systems; however, this does not imply the exclusion of domain experts or the adoption of a uniform approach in every context. In critical fields such as healthcare and aviation, for instance, broad participation must be balanced with deep technical expertise to maintain safety and reliability. In these high-stakes domains, effective citizen control may take a hybrid form, wherein communities shape overall values and objectives while specialists guide specific technical parameters. Thus, citizen sovereignty in AI does not preclude expert collaboration; rather, it enables stakeholders to determine when and how specialized knowledge interacts with collective oversight.

Figure 2 illustrates how agency, transparency, inclusivity, and governance structures vary across these tiers. This

progression also highlights the transition from instrumental consumerism to communal sovereignty, guiding evaluations of existing approaches and helping chart paths toward more participatory paradigms.

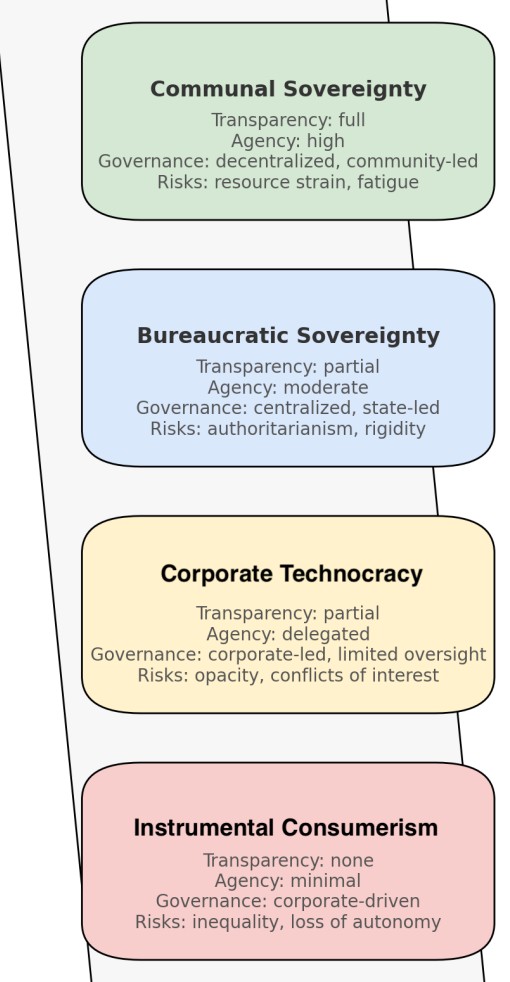

*Figure 2.* Progression in stakeholder power from minimal engagement (Consumer-Based) to robust self-governance (Communal Sovereignty). This categorization helps assess current initiatives and guide transitions toward more participatory models.

While our four-tier model is inspired by Arnstein's seminal ladder of participation, it introduces AI-specific mechanisms for transitioning between tiers. For instance, the shift from *Consumer-Based* to *Private Organization-Led* may occur when communities adopt structured feedback channels and partner with companies to revise product roadmaps or data-sharing agreements. A subsequent transition to the *Government-Controlled* tier might involve formal policy mandates requiring community audits or the creation of statutory councils with partial decision-making authority over AI deployments. Finally, the *Citizen-Controlled* tier relies on advanced institutional support—such as local or cooperative data trusts with legally enforceable ownership

structures and educational programs that foster the technical competence necessary for meaningful oversight.

At each transition point, stakeholder roles evolve—from reacting to system outputs, to co-managing data and design objectives, to exercising decisive authority over governance processes. By outlining these transitions, we aim to illustrate how communities can incrementally acquire the capacity to shape AI systems, moving beyond surface-level consultation or feedback.

## 6. Lessons from Participatory Practices

**The extent to which participatory AI can reconfigure decision-making power—or instead uphold existing technological agendas—remains contested.** This section draws on empirical insights from a range of participatory AI initiatives (Table 1) to explore whether stakeholder engagement can meaningfully influence AI design or remains largely symbolic. Although many projects prioritize knowledge sharing rather than deeper power-sharing, they also reveal both the opportunities and constraints that shape more robust forms of community involvement.

Participatory AI initiatives in education, healthcare, urban planning, and software development often involve stakeholders with direct interests in AI-driven decisions (Jacobs, 1961; Lee et al., 2019; Zicari et al., 2021; Zhang & Aslan, 2021; Sieber et al., 2024b). For example, *Co-Design of Trustworthy AI in Healthcare* (Zicari et al., 2021) included patients, clinicians, and ethicists to expose biases in diagnostic tools, leading to enhanced accountability despite resource and expertise challenges. Urban planning efforts, such as *MID-Space* (Nayak et al., 2024), relied on iterative community annotation and mediation to address conflicting priorities (Sloane et al., 2022). These examples highlight both the promise of stakeholder inclusion and the structural, institutional, and practical barriers that may limit its impact.

Across the nine case studies, success factors consistently included the early and sustained involvement of local stakeholders (e.g., language experts, community volunteers), transparent articulation of goals and benefits, and explicit acknowledgment of resource inequalities. For example, in the WeBuildAI case study (Lee et al., 2019), the framework enabled diverse participants—including donors, volunteers, recipient organizations, and staff—to collaboratively design a matching algorithm for donation allocation. Evaluation using historical data revealed that the algorithm produced more equitable outcomes than traditional human-led methods. Specifically, it reallocated donations in ways that better prioritized organizations serving communities with higher poverty rates, lower incomes, and limited food access. Participants reported increased trust in, and clearer understanding of, the algorithmic decision-making process.

*Table 1.* Nine Examples of Participatory AI

| Project | Why It Was Done | How It Was Implemented | Stakeholder Involvement | Domain / Application | Key Outcomes & Impact |
|---|---|---|---|---|---|
| **Anthropic's Collective Constitutional AI (Huang et al., 2024b)** | Align AI with shared values | Ethical constitution, iterative feedback | AI researchers, end-users, ethicists | AI alignment | Exposed tensions in ethical frameworks |
| **PRISM Alignment Dataset (Kirk et al., 2024)** | Investigate cross-cultural alignment | Surveys of 1,500 participants | International participants, researchers | AI ethics | Revealed cultural disagreements |
| **MID-Space (Nayak et al., 2024)** | Democratize design visualization | Community-based annotation | Marginalized groups, planners | Urban planning | Incorporated localized perspectives |
| **Participatory Modelling for Agro-Pastoral Restoration (Eitzel et al., 2021)** | Include Indigenous knowledge | Co-created computational models | Farmers, modelers | Environmental sustainability | Context-driven land management solutions |
| **Co-Design of Trustworthy AI in Healthcare (Zicari et al., 2021)** | Address bias in medical AI | Iterative design with patients, clinicians | Patients, ethicists | Healthcare | Reduced diagnostic bias, enhanced trust |
| **Project Dorian (Berditchevskaia et al., 2021)** | Adapt AI for humanitarian settings | Human-in-the-loop feedback | NGO staff, data scientists | Crisis logistics | Facilitated faster resource allocation |
| **WeBuildAI: Participatory Algorithmic Governance (Lee et al., 2019)** | Develop collaborative governance | Workshops with civic groups | Civic groups, public officials | Computer science | Prototype participatory algorithms |
| **Participatory Research for Low-resourced Machine Translation (Nekoto et al., 2020)** | Scale NLP for low-resource African languages | Community-driven data collection, annotation, and workshops | African language speakers, researchers, linguists | Machine Translation, NLP | Novel datasets and benchmarks for over 30 languages; enabled community contributions |
| **Māori Data Sovereignty Initiative (Kukutai & Taylor, 2016)** | Protect Māori language data and ensure community benefits | Establish Māori Data Sovereignty Protocols, community-led annotation | Māori community, linguists, indigenous organizations | Language technology, data sovereignty | Controlled data sharing, preservation of autonomy, community-led tech development |

**Early Engagement** Several projects, including *Participatory Modelling for Agro-Pastoral Restoration* and *PRISM Alignment Dataset*, show that engaging communities early can reveal cultural or ethical issues before they become entrenched. Delayed consultation often feels tokenistic, limiting participants' ability to influence fundamental design decisions (Arnstein, 1969; Huang et al., 2024b;a).

**Conflict Resolution and Power Dynamics** Differences in moral frameworks or cultural norms may create tensions if not managed proactively. For example, *PRISM Alignment Dataset* (Kirk et al., 2024) identified cross-cultural disagreements about AI ethics. Resource imbalances can also permit well-funded institutions to dominate agenda-setting, marginalizing other voices (Benthall & Haynes, 2019; Eitzel et al., 2021; Cachat-Rosset & Klarsfeld, 2023;

Murgia, 2024; Ulnicane, 2024).

**Resource Commitments** Several initiatives, including *Māori Data Sovereignty Initiative* and *Project Dorian*, relied on training, funding, and organizational support (Kukutai & Taylor, 2016; Berditchevskaia et al., 2021). Communities that cannot independently access these resources may depend on external programs that come with distinct priorities (Tilmes, 2022; Birhane et al., 2022; Gerdes, 2022; Sloane et al., 2022).

**Conflation and Cooptation** While increasing the number of participants can broaden representation, it does not necessarily equate to deeper engagement (Bohman, 2000; Birhane et al., 2022). Companies may harness grassroots involvement mainly for publicity or profit, reframing local

knowledge for external gain (Mikalef et al., 2022; Murgia, 2024). For instance, in African language machine translation (Nekoto et al., 2020), some community efforts have been repackaged as commodifiable assets by external actors.

**Balancing Expert and Local Knowledge**  Efforts to integrate specialized and local knowledge can face disagreements over data validity, model interpretability, and ethical guidelines (Birhane et al., 2022). Translational strategies—ranging from interdisciplinary facilitation teams to community-guided metrics—can help mediate these gaps (Fischer, 2000; Lee et al., 2021).

**Implications for a Right to AI**  While most cases in Table 1 are context-specific, some show that sustained grassroots advocacy can influence decision-making. Jane Jacobs' fight against highway expansion (Jacobs, 1961) highlights how informed stakeholders and activism shape planning. In AI, persistent public pressure could counter performative engagement. Early participatory methods that grant real decision-making power are more likely to redistribute power, benefiting broader communities.

## 7. Recommendations

To effectively realize the Right to AI, the following recommendations are designed as a collaborative, multi-sectoral effort. Recognizing that structural change cannot be achieved by a single stakeholder alone, these proposals engage a diverse range of actors—including educational institutions, governmental bodies, community organizations, and industry partners—to work together in fostering ethical, accountable, and inclusive AI systems.

**Provide Technical and Educational Resources**  Organizations such as universities, NGOs, and local governments can collaborate to develop workshops, open educational materials, or interactive simulators that demystify AI. These efforts equip community members, public officials, and civic groups with foundational AI knowledge, enabling them to question design choices, scrutinize potential risks, and hold system implementers accountable (Almatrafi et al., 2024). If these initiatives remain underfunded or absent, communities may lack the means to exercise meaningful oversight.

**Facilitate Participation**  Developers, civic tech groups, and service providers may deploy accessible interfaces—such as real-time translations or interactive dashboards—to broaden engagement in AI projects (Anthropic, 2023; Williams et al., 2024; Sieber et al., 2024b). Large language models allow code-free inclusive interfaces, enabling broader participation in AI governance and design. Structured feedback and co-creation sessions encourage non-experts to contribute insights into model objectives or

flagged decisions (Huang et al., 2024a). If these methods are neglected, only a narrow segment of technically proficient stakeholders may shape AI systems.

**Formalize Community Assemblies**  Municipalities, civic groups, and industry partners can establish local AI councils with advisory roles (Bohman, 2000). Over time, these bodies may gain decision-making authority, ensuring public influence on AI-driven processes and preventing ethical or societal oversights.

**Establish Data Trusts and Auditing Processes**  Governments, philanthropies, and private-sector coalitions can create community-based data trusts to govern training data, consent, and benefit distribution (Sieber et al., 2024b; Birhane et al., 2022). Transparent auditing—accessible to both laypersons and experts—would enhance accountability and prevent unchecked data abuses (Zaidan & Ibrahim, 2024).

**Localized Adaptation**  Local AI developers, community organizations, and domain experts can fine-tune generative models with smaller, context-specific datasets (Nayak et al., 2024; Kirk et al., 2024). By involving residents or practitioners in curation and training, these models can better reflect local norms and languages (Mishra, 2023). Failure to integrate local context risks producing irrelevant or culturally misaligned AI outputs, weakening public trust and engagement (Huang et al., 2024a).

**Integrate Conflict Resolution and Mediation**  Policymakers, community leaders, and mediators can establish transparent panels to address ethical disputes, stakeholder conflicts, and cultural sensitivities (Femia, 1996; Bondi et al., 2021). These panels balance technical feasibility with social imperatives, fostering trust in AI governance. Without them, unresolved conflicts may deter community participation and reinforce power imbalances.

**Mobilize Researchers for Community Engagement**  As machine learning researchers are well equipped to raise awareness and morally support their surrounding communities through communication and dialogue about AI, this responsibility translates into practical steps. **Implementing the Right to AI would prompt researchers to engage more systematically with non-technical stakeholders.** This could involve structuring datasets with transparent documentation, designing interfaces for community feedback, and integrating diverse perspectives into model objectives. Such a shift may introduce time and resource overheads. However, it also alters the dynamics of accountability and offers opportunities to mitigate biases and strengthen public trust. Balancing technical efficiency with meaningful public engagement may require interdisciplinary collaboration, new skill sets—such as facilitation—and iterative design

cycles. Although demanding, this participatory approach can enhance the relevance and robustness of AI systems while reinforcing public confidence in their development.

**Bridge Technical Gaps**   Beyond political and normative considerations, substantial technical gaps remain in realizing the Right to AI across all tiers. Participatory systems must offer accessible, language-inclusive interfaces that accommodate diverse forms of engagement and varying abilities, without oversimplifying complex modeling decisions. Moreover, conflict-resolution protocols require both computational and sociotechnical research to systematically integrate diverse perspectives into model design—shifting from disaggregated to aggregated viewpoints in a transparent and traceable manner. The development of explainability and interpretability tools tailored to non-experts remains in its early stages. Finally, reliable methods are needed to validate the quality and relevance of community-contributed data, particularly in regions with limited technical capacity. Addressing these challenges is essential to ensure that the Right to AI becomes not merely aspirational, but operationalized in a sustainable and equitable manner.

## 8. Alternative Views

Some scholars and practitioners question whether broad-based participatory approaches to AI are feasible or desirable. A market-led perspective asserts that competition and consumer choice will naturally drive responsible AI (Dignam, 2020; de Marcellis-Warin et al., 2022; Hadfield & Clark, 2023; Judge et al., 2024), though such models can overlook communities lacking purchasing power or market influence (André et al., 2018; Radu, 2021; Cohen & Suzor, 2024; Ulnicane, 2024). The Right to AI maintains that these gaps warrant structured stakeholder participation to include marginalized voices and address power asymmetries.

Others emphasize strong state oversight to ensure consistent regulation and enforcement (de Almeida et al., 2021; Schmitt, 2022; Bengio et al., 2024). Critics of localized governance argue that citizen bodies may lack the necessary expertise, risk fragmentation, or amplify parochial biases (de Almeida et al., 2021; Murgia, 2024; Shepardson et al., 2024). In contrast, the Right to AI can complement centralized regulation through decentralized governance, enabling community-specific adaptations while maintaining broad standards. Carefully designed conflict-resolution methods can limit local biases and encourage inclusive decision-making.

In sum, we do not suggest that participatory governance alone can resolve all problems of AI oversight. Rather, the Right to AI offers a missing dimension—namely, inclusive, community-driven frameworks that complement both market incentives and centralized regulations.

## 9. Conclusion

The widespread adoption of AI raises questions about democratic oversight, social justice, and epistemic diversity. This paper proposes a Right to AI, aiming to shift from expert-dominated decision-making toward participatory approaches in which communities influence how AI infrastructure is designed, deployed, and governed. Drawing on Lefebvre's *Right to the City* and Arnstein's ladder of participation, the argument suggests viewing AI as societal infrastructure that requires sustained and inclusive governance.

Case studies were examined to demonstrate the potential and challenges of participatory efforts, highlighting issues such as resource inequalities, value pluralism, and institutional inertia. Recommendations, including structured community assemblies, data trusts, iterative governance, and conflict mediation, were outlined to operationalize the Right to AI. These measures aim to ensure that AI systems reflect community values, address biases, and preserve autonomy.

The paper contends that the Right to AI is an important component of the future AI ecosystems because it addresses the interplay of autonomy, trust, and accountability in technology development. Advancing this right entails collective learning, institutional innovation, and ongoing negotiation of values among diverse constituencies. As AI continues to influence educational curricula, medical diagnostics, economic opportunities, and civic engagement, the need for inclusive governance increases.

Future research can expand the philosophical and practical foundations of the Right to AI, supporting its necessity and details of its implementation. Such work might include a thorough examination of the four-tier ladder model—which conceptualizes the four modes in which participatory AI is practiced—challenging existing frameworks by aligning AI governance with these tiers. Scholars can explore methods to mobilize citizens under the Right to AI umbrella, fostering widespread engagement and ensuring that participatory governance mechanisms are inclusive and representative. Integrating interdisciplinary perspectives from political theory, ethics, and technology studies can also serve to enhance the grounding and reasoning for the Right to AI. Adapting the four-tier ladder to diverse cultural contexts is crucial for scalability and global implementation, while addressing feasibility and funding—determining who pays and how to sustain participatory mechanisms—is equally essential.

By addressing these areas, research can develop the Right to AI as a comprehensive framework that aligns technical advancements with communal interests, promoting inclusivity, transparency, and ethical accountability in AI governance.

## Impact Statement

By reframing AI as societal infrastructure and proposing a *Right to AI*, this work highlights both opportunities and challenges for democratizing AI governance. The benefit of participatory governance is that it can address biases in data and model design, empower historically marginalized communities, and foster trust in AI systems. By asserting collective ownership of data, citizen control of model objectives, and localized adaptation, the proposed framework encourages equitable distributions of AI's benefits, aligning technological progress with diverse cultural and ethical perspectives.

However, realizing such a participatory model may face significant hurdles in practice. Communities may lack the educational resources or institutional support to contribute meaningfully, and power asymmetries could lead to tokenistic engagement rather than genuine reform. Additionally, expanding citizen-led decision-making could risk misaligned local biases or fragmented national regulations if not carefully mediated. Nevertheless, by centering the voices of those most affected by AI, the *Right to AI* offers a transformative path that can mitigate systemic biases and strengthen democratic ideals in an increasingly automated world.

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

# Appendix

- **Consolidated Overview**

- **A Governance Right**

- **B Implementation Path**

- **C Generative Agents**

- **D Power & Data**

- **E Ethical Grounds**

- **F Hidden Choices**

## Consolidated Overview

Contemporary discourse on AI governance encompasses a broad spectrum of policy proposals, ethical guidelines, and technical approaches aimed at aligning AI systems with societal values (Mishra, 2023; Zaidan & Ibrahim, 2024; Sorensen et al., 2024). Major institutions such as the OECD and the European Union have introduced frameworks for responsible AI development, often emphasizing fairness, accountability, and transparency (Saheb & Saheb, 2024; Zhang et al., 2025). However, these initiatives vary widely in design and enforceability across different regions.

For instance, while the European Union's proposed AI Act imposes legally binding obligations for high-risk AI systems (European Union, 2024), other jurisdictions frequently rely on voluntary guidelines or market-driven standards (Kerry et al., 2025). In the United States, the now-rescinded White House Blueprint for an AI Bill of Rights was primarily composed of non-binding principles, and no comprehensive replacement has emerged (The White House, 2025). In Asia, countries such as Singapore emphasize industry consultation and codes of practice, whereas Japan's guidelines seek to harmonize technological innovation with broader societal goals (Kerry et al., 2025).

Several developing countries have drafted national AI strategies or joined initiatives like the African Union's Continental Strategy for AI, but often face structural and financial barriers that constrain robust public participation (Bernstein & Bekheit, 2024). Overall, these disparate regulatory models illustrate an ongoing tension between top-down or expert-led approaches and a growing demand for more inclusive governance. The Right to AI advanced in this paper seeks to bridge these gaps by advocating participatory frameworks, human-centered design, and collective data ownership, thus complementing rather than merely supplanting existing governance structures across diverse legal and cultural contexts.

This paper's core argument can be summarized in three steps: first, that AI increasingly functions as a form of societal infrastructure; second, that individuals and communities hold a right to help shape and govern the infrastructure that affects their lives (Benkler, 2006); and third, that therefore a "Right to AI" naturally follows. As elaborated in the main text, Section 3 advances the conceptual

framing of AI as infrastructure, while Section 4 outlines four normative grounds—ethical, political, epistemic, and institutional—that justify this right.

In brief, we argue that:

1. to protect democratic legitimacy, impacted communities must have meaningful decision-making power over AI design;

2. social justice demands attention to marginalized voices;

3. epistemic autonomy requires safeguards against narrow or monolithic knowledge curation by AI;

4. data, which is essential for AI, is socially produced and thus merits communal oversight.

In grounding these claims, our framework views AI as not merely a product of private innovation but an infrastructure of public consequence, and it aligns with global calls for more inclusive and participatory technological governance (Crawford, 2021).

One might wonder if the Right to AI merely collapses into a broader right to infrastructure. While it does share foundational principles with other governance rights over public goods, AI's particular characteristics—such as algorithmic opacity, global data reliance, and evolving automation capabilities—warrant a dedicated framework (Cohen, 2019). The Right to AI thus refines a general right to shape infrastructure, specifying the necessary mechanisms to address the unique ethical, technical, and political complexities posed by AI.

Some scholars and industry leaders dispute the premise that AI is or should be treated as public infrastructure, arguing that AI's intangibility sets it apart from utilities like water or electricity. Others emphasize market-led solutions, suggesting that competition will naturally encourage responsible AI. However, the infrastructural viewpoint spotlights the breadth and depth of AI's societal reach, from healthcare to education to political discourse (Plantin et al., 2018). This, we contend, justifies a governance paradigm akin to that of publicly regulated utilities. Treating AI as infrastructure thus reframes oversight as a collective undertaking, rather than a question of consumer choice or proprietary rights alone.

## A. Governance Right

The Right to AI can be understood as a *governance right*, emphasizing *policy*, *procedural justice*, and *institutional design* (Habermas, 1996; Ostrom, 1996). Rather than relying on market mechanisms or top-down state control, gover-

nance rights establish frameworks through which *individuals and communities* can co-determine AI systems' objectives and oversight structures. This perspective draws on democratic traditions recognizing the capacity of the public to influence technological developments that shape collective well-being (Sun, 2020; Wenar, 2023).

In this framework, the Right to AI moves beyond a *privilege right*—which might only allow people to use a given technology—to a *power right*, which grants communities the authority to *reshape* AI systems (Sun, 2020; Wenar, 2023). While intellectual property laws may protect patents or licenses, the broader direction, governance, and deployment of AI can be subject to public deliberation. Examples include local AI assemblies, public audits, and cooperative data stewardship, each aiming to reconcile private ownership with communal oversight (Lee et al., 2021; Schiff et al., 2021).

## B. Implementation Path

**(a) Empirical Evidence from Participatory AI** Initiatives such as *WeBuildAI* (Lee et al., 2019) and *MID-Space* (Nayak et al., 2024) suggest that community participation can align algorithmic outputs with local values. These projects have found that when participants understand how and why certain data are used, they are more inclined to trust and engage with AI tools. However, repeated consultations without tangible outcomes may cause *participation fatigue* (Arnstein, 1969).

**(b) Evaluation of Participatory Models** Comparative analyses of participatory and non-participatory AI systems could measure outcomes such as transparency, fairness, and community trust (Huang et al., 2024b; Kirk et al., 2024). Involving domain experts, local knowledge holders, and impacted communities may help refine evaluation criteria and metrics (Birhane et al., 2022; Sloane et al., 2022).

**(c) Technical Approaches** Methods like *Reinforcement Learning from Human Feedback* (RLHF) (Bai et al., 2022a) and *participatory fine-tuning* (Kirk et al., 2024) enable stakeholder input on model behaviors. Balancing diverse viewpoints in these processes can be challenging but may be facilitated by transparent data pipelines and iterative design cycles (Anthropic, 2023; Birhane et al., 2022).

**(d) Scalability and Institutional Barriers** Scaling participatory approaches to national or international contexts is complex. Bureaucratic structures and profit-driven goals sometimes dilute community-driven decision-making (Huang et al., 2024a). Hybrid frameworks that combine local autonomy with standardized guidelines might help retain the participatory ethos (Sieber et al., 2024a).

**(e) Applications Across System Types**   Participatory governance can apply to various AI domains but may face context-specific constraints. For instance, specialized knowledge or resource limitations can limit who can engage. Below are select examples:

**(e.1) Education and Healthcare**   End-users often have immediate stakes in these areas (Zicari et al., 2021; Zhang & Aslan, 2021). Collaborative tools have been piloted to identify biases in diagnostic algorithms (Zicari et al., 2021), though sustained adoption can require institutional support and specialized expertise.

**(e.2) Urban Planning**   Urban planning regularly involves public input, though execution can vary (Jacobs, 1961; Sieber et al., 2024b). Projects like *MID-Space* used iterative community annotation to inform planning tools (Nayak et al., 2024), revealing how structured feedback loops might help integrate diverse local needs (Mushkani et al., 2025b).

**(e.3) Software Development**   Open-source and agile methods stress iterative engagement. *WeBuildAI* (Lee et al., 2019) involved workshops where participants shaped algorithmic governance. Transparent norms and distributed authority appeared pivotal to maintaining motivation and commitment.

**(f) Future Research**   Further areas of inquiry include:

- *Data Practices and Local Expertise*: Co-created annotation and Indigenous knowledge integration may enhance system credibility (Eitzel et al., 2021; Nayak et al., 2024).

- *Longitudinal Studies*: Investigating how participation evolves over time, focusing on trust-building and avoiding *participation fatigue* (Sloane et al., 2022).

- *Sustainability*: Allocating resources to ensure consistent engagement and demonstrate visible influence on policy or system outputs (Ulnicane, 2024).

## C. Generative Agents

Recent advances in large language models and other generative systems allow for large-scale content creation across text, images, or interactive dialogues (Bommasani et al., 2022; Lazar, 2024). Several factors may benefit from participatory governance:

**Pluralistic Alignment**   Generative AI can reinforce majority perspectives if minority viewpoints are underrepresented in the training data (Bai et al., 2022a; Huang et al., 2024a; Sorensen et al., 2024). Approaches like RLHF may not fully capture diverse views, prompting research on methods such as Overton pluralism or jury-based alignment (Sorensen et al., 2024). These efforts could mitigate homogenization of perspectives and enhance equitable representation (Huang et al., 2024b).

**Risk of Amplified Disinformation**   Generative models may facilitate the rapid creation of misleading or harmful content (Tenove et al., 2018; Zhang et al., 2025). While community monitoring and co-governance can assist in mitigating such content, institutional safeguards and digital literacy programs may be crucial for broader resilience (Zhou et al., 2024).

**Data Transparency and Ownership**   Large-scale data scraping is central to many generative systems (Kukutai & Taylor, 2016; Miller, 2019; Kitchin, 2023). A Right to AI perspective could motivate community-based decisions about data collection, retention, and licensing (Kukutai & Taylor, 2016).

**Algorithmic Profiling and Manipulation**   Adaptive agents can generate detailed user profiles by monitoring interactions, raising concerns over targeted manipulation or preferential targeting (Leike et al., 2018; Ray, 2023; Kitchin, 2023). Participatory audits and interpretability tools might help users and regulators detect problematic patterns, but effective governance likely requires ongoing transparency about model objectives (Birhane et al., 2022; Huang et al., 2024a).

## D. Power & Data

A Foucauldian perspective suggests that *marginalization and exclusion* often result from institutional power relations and discursive frameworks that limit whose voices are considered legitimate (Foucault, 1975). In AI, control over design, deployment, and data policies can be concentrated among corporations or governmental actors. Changing these power structures may require new or revised institutional processes that invite broader participation.

**Article 27 of the UDHR and the Right to Science**   Article 27 of the *Universal Declaration of Human Rights (1948)* states that everyone should share in "scientific advancement and its benefits." Contemporary interpretations extend this to digital and technological domains (Sun, 2020; Wenar, 2023). However, public accessibility of data does not necessarily translate to equitable involvement in systems built upon it. Proprietary protections can confine tangible benefits to a limited number of stakeholders.

**Data Enclosure**   Some private actors train AI models on publicly available data and then restrict or monetize the results, a process sometimes referred to as *data enclosure*

(Beer, 2016; Kitchin, 2016). Critics argue that in fields such as healthcare and policing, models used without public oversight can exacerbate social inequalities (Eubanks, 2018; Avellan et al., 2020). The Right to AI positions communities to scrutinize and influence such models, aiming to prevent the privatization of communal knowledge.

## E. Ethical Grounds

**Respect for Moral Agency**   A fundamental argument for the Right to AI is grounded in respect for moral agency. AI systems significantly influence people's lives, making decisions on employment, healthcare, policing, and education (Eidelson, 2015; Laitinen & Sahlgren, 2021; Mackenzie, 2015). Ensuring that individuals have a role in shaping these systems aligns with principles of autonomy and self-determination (Stoljar, 2014; Tasioulas, 2023). Without participatory engagement, AI risks reducing individuals to passive subjects of algorithmic governance rather than active contributors to its development.

**Control Over Personal Information**   AI-based decisions often rely on personal data, prompting questions about privacy, consent, and user control (Marmor, 2015; Wachter & Mittelstadt, 2019). Mechanisms embedded in the Right to AI could clarify data handling processes and reduce unwarranted intrusions (Rumbold & Wilson, 2019; Floridi, 2014).

**Mitigating Intrusion and Anonymization Risks**   AI's ability to infer personal attributes, even when explicit data is not provided, raises serious ethical concerns (Mühlhoff, 2023; Henley, 2021). These risks can be mitigated through participatory oversight mechanisms, ensuring that AI does not perpetuate intrusive or harmful data practices (Burrell, 2016; Loi & Christen, 2020). By advocating for the Right to AI, communities can establish consent-based frameworks that prioritize ethical data handling.

**Addressing Statistical Discrimination**   AI often relies on statistical generalizations that may fail to respect the uniqueness of individuals (Adams-Prassl et al., 2023; Barocas & Selbst, 2016; Kleinberg et al., 2019). A participatory approach to AI governance would enable affected communities to challenge harmful biases and demand equitable algorithmic design (Harcourt, 2007; Larson et al., 2016). The Right to AI provides a means for individuals to contest algorithmic categorizations and push for more inclusive and fair outcomes (Chen, 2023).

**Obligation to Provide Justification**   When AI influences critical life decisions, increased transparency and explainability may be warranted (Grant et al., 2023; Vredenburgh, 2022; Rubel et al., 2020; Tasioulas, 2023). A Right to AI

approach aligns with the notion that those subject to algorithmic decisions should have a means to access comprehensible justifications (Candrian & Scherer, 2022).

## F. Hidden Choices

We end this paper with below analogy:

**Imagine it's 2035...**

You walk into a restaurant, but you don't order—your meal has already been decided for you. The chefs claim to know your tastes, preferences, and needs better than you do. The recipes are hidden, the kitchen is closed to outsiders, and any attempt to question or change your meal is met with silence. If this is the only restaurant in town, your choices aren't just limited—they're non-existant.

Now, imagine AI works in the same way. A small group of actors dictate what information you see and which services you can access. The *Right to AI* challenges this imbalance, asserting that communities should not merely be passive consumers but active participants in designing, governing, and overseeing AI. To maintain our autonomy and choice, we must have a say in how the AI that dictates our preferences and choices is built and deployed.

