# OpenReview forum: "Position: The Right to AI"
_ICML.cc/2025/Position_Paper_Track — ICML 2025 Position Paper Track poster_

### Official Review · Reviewer_2K8h · 2025-02-17

**Significance:** 3
**Argument Clarity:** 3
**Ethics Flag:** Yes
**Rating:** 3
**Confidence:** 4

**Questions:**

How would the proposed participatory mechanisms handle scenarios where different stakeholder groups have fundamentally conflicting values or priorities?

What minimal level of technical literacy would be required for meaningful participation in AI governance?

How would the proposed framework handle rapidly evolving AI capabilities that may outpace traditional participatory processes?

**Discussion Potential:**

4

**Ethics Review Area:**

["Discrimination / Bias / Fairness Concerns", "Inappropriate Potential Applications & Impact (e.g., human rights concerns)"]

**Paper Summary:**

This position paper proposes and argues for establishing a "Right to AI" as a framework for participatory governance of AI systems. Drawing on Henri Lefebvre's "Right to the City" concept - nice to see him cited within the AI academic sphere - it argues for reconceptualizing AI as societal infrastructure rather than just a product of expert design. The paper develops a four-tier model for implementing this right, ranging from minimal consumer-based engagement to full citizen control. It analyzes nine case studies of participatory AI initiatives and provides recommendations for practical implementation

**Position:**

Yes

**Position In Title:**

Yes

**Related Work:**

2

**Strengths And Weaknesses:**

1. Core Issues with Contribution and Novelty
The paper's primary weakness is that it largely synthesizes existing work on AI governance and participation without providing substantive new theoretical frameworks or empirical validation. While it draws interesting parallels between AI governance and urban planning through Lefebvre's "Right to the City" concept, it does not sufficiently develop this analogy into a novel theoretical contribution.
The four-tier model presented in Section 5 ("Consumer-Based", "Private Organization-Led", "Government-Controlled", and "Citizen-Controlled") appears to be primarily a reframing of Arnstein's existing ladder of participation rather than a meaningful theoretical advancement. The paper does not rigorously define the relationships between these tiers or provide concrete mechanisms for transitioning between them.

2. Methodological and Empirical Limitations
While the paper references nine case studies in Table 1, it does not provide systematic analysis of these cases. The discussion remains largely at a surface level, without detailed examination of success factors, failure modes, or generalizable insights. For example, when discussing WeBuildAI and MID-Space projects, the paper notes they "suggest that community participation can align algorithmic outputs with local values" but does not provide specific evidence or metrics for this alignment... the paper makes broad claims about participatory governance improving AI outcomes without providing concrete evidence. For instance, the statement "grassroots participatory methodologies can mitigate biased outcomes and enhance social responsiveness" (abstract) is not supported by empirical validation or specific mechanisms.

3. Theoretical Framework Weaknesses
The theoretical foundation attempting to link democratic theory, social justice, and data governance remains underdeveloped. The paper introduces multiple theoretical concepts (e.g., epistemic autonomy, democratic legitimacy) without rigorously defining their relationships or providing a cohesive framework for analysis. the engagement with critical technical concepts like "algorithmic bias" and "model interpretability" remains superficial. For example, when discussing technical approaches in Appendix B, it mentions "Methods like Reinforcement Learning from Human Feedback (RLHF)" without examining the specific challenges and limitations of applying RLHF in participatory contexts.

4. Implementation and Practicality Gaps
The recommendations section lacks concrete guidance for implementing the proposed Right to AI. While it suggests mechanisms like "local AI councils" and "public audits," it does not address critical practical challenges such as:
- Required technical expertise for meaningful participation
- Resource allocation and sustainability
- Reconciliation of conflicting stakeholder interests
- Integration with existing regulatory frameworks

To enhance the position's support, the authors could more explicitly address scalability challenges in participatory governance and provide more concrete metrics for evaluating successful implementation

The paper raises important issues about AI governance but requires substantial revision to meet the standards for a major conference contribution. As currently written, it reads more as a position statement than a scholarly advancement of the field. These issues appear fundamental enough that major revision would be required for the paper to make a substantive contribution

**Support:**

3

---

> ### Author Rebuttal · Authors · 2025-04-01
>
> Thank you to the reviewer for the thoughtful comments. We address the concerns on novelty and related questions below.
>
> ---
> ### 1. Core issues: limited contribution, unclear novelty, missing tier links and transitions.
>
> **Response:**
> The paper extends Lefebvre’s framework to AI through the concept of a “Right to AI,” proposing a four-tier governance model that treats AI as shared societal infrastructure. Drawing on Arnstein’s ladder, it outlines how communities can move from minimal to full control, supported by concrete mechanisms such as local AI councils, data trusts, and participatory audits. This synthesis offers a coherent and original model for collective AI oversight that goes beyond existing analogies.
>
> We provide the following to highlight our contribution.
>
> **Added paragraph in section 5:**
> *While our four-tier model is inspired by Arnstein’s seminal ladder of participation, it introduces AI-specific mechanisms for transitioning between tiers. For instance, the shift from “Consumer-Based” to “Private Organization-Led” may occur when communities adopt structured feedback channels and partner with companies to revise product roadmaps or data-sharing agreements. A subsequent transition to the “Government-Controlled” tier might involve formal policy mandates requiring community audits or the creation of statutory councils with partial decision-making authority over AI deployments. Finally, the “Citizen-Controlled” tier relies on advanced institutional support—such as local or cooperative data trusts with legally enforceable ownership structures and educational programs that foster the technical competence necessary for meaningful oversight. At each transition point, stakeholder roles evolve—from reacting to system outputs, to co-managing data and design objectives, to exercising decisive authority over governance processes. By outlining these transitions, we aim to illustrate how communities can incrementally acquire the capacity to shape AI systems, moving beyond surface-level consultation or feedback.*
>
> ---
> ### 2. Methodological and empirical limitations
>
> **Response:**
> We now include an expanded synthesis of the cases presented in Table 1 (available in the Appendix), highlighting key participation outcomes, resource constraints, and observed shifts in power. To address the concern that "the paper makes broad claims about participatory governance improving AI outcomes without providing concrete evidence," we draw on insights from our nine case studies, as illustrated below.
>
> **Revised paragraphs in Section 6:**
> *Across the nine case studies, success factors consistently included the early and sustained involvement of local stakeholders (e.g., language experts, community volunteers), transparent articulation of goals and benefits, and explicit acknowledgment of resource inequalities. For example, in the WeBuildAI case study (Lee et al., 2019), the framework enabled diverse participants—including donors, volunteers, recipient organizations, and staff—to collaboratively design a matching algorithm for donation allocation. Evaluation using historical data revealed that the algorithm produced more equitable outcomes than traditional human-led methods. Specifically, it reallocated donations in ways that better prioritized organizations serving communities with higher poverty rates, lower incomes, and limited food access. Participants reported increased trust in, and clearer understanding of, the algorithmic decision-making process.*
>
> ---
> ### 3. Implementation and practicality gaps
>
> By discussing, acknowledging, and advocating for these concerns, we have outlined the barriers to realizing the Right to AI in the conclusion section. We have also addressed this point in our responses to reviewers Gwpg (Comment W3: how the Right to AI impacts ML researchers) and Q58X (Comments 1, 2, and 5).
>
> ---
>
> Regarding value conflicts, the paper argues that inclusive participation enables all groups to voice their interests. While detailed resolution—through deliberative platforms or mediation—is beyond this paper’s scope, focused on the right to advocate, we address it in the Appendix.
>
> On the matter of technical literacy, our proposal affirms that all those affected by AI deserve a direct or delegated voice, regardless of technical expertise, and calls for accessible tools and education to support meaningful participation.
>
> Regarding rapidly evolving AI capabilities, we frame the Right to AI as a response to accelerating technological change. Participatory governance structures—such as iterative citizen assemblies or adaptive regulatory boards—offer mechanisms for continuous oversight.
>
> ---
> **We appreciate your insights and have revised the manuscript to clarify our theoretical contributions, broaden methodological considerations, and strengthen practical recommendations. Our proposal offers a timely perspective on AI governance, aiming to spark ongoing inquiry rather than finalize all implementation details.**

---

> > ### Comment · Reviewer_2K8h · 2025-04-03
> >
> > I acknowledge the authors' efforts to address my concerns. Their rebuttal indicates planned improvements to: (1) clarify tier relationships in their model with concrete transition mechanisms; (2) strengthen case study analysis with empirical outcomes from WeBuildAI showing improved equity metrics; (3) expand theoretical connections between AI governance and participatory frameworks; and (4) add implementation guidance in the conclusion.
> > However, I remain concerned that the fundamental issue of novelty persists. While the paper offers a valuable synthesis of participatory governance concepts applied to AI, the proposed "Right to AI" framework still appears to be primarily a reframing of existing concepts rather than a substantial theoretical advancement. The methodological approach to case studies, while improved with additional detail, continues to lack systematic analysis.
> >
> > I changed to 3: Weak accept

---

### Official Review · Reviewer_Gwpg · 2025-02-22

**Significance:** 2
**Argument Clarity:** 2
**Rating:** 3
**Confidence:** 3

**Questions:**

Please see the weaknesses.

**Discussion Potential:**

3

**Paper Summary:**

This paper introduces the concept of a *"Right to AI”*, framing AI as a societal infrastructure and arguing that individuals and communities affected by AI systems should have the capacity and entitlement to shape, critique, and govern these infrastructures. Drawing inspiration from the Right to the City in urban theory, the authors propose recommendations to implement the proposed Right based on the Ladder of Right to AI.

**Position:**

Yes

**Position In Title:**

Yes

**Related Work:**

3

**Strengths And Weaknesses:**

**S1**: The paper is well-structured and clearly written, with a logical flow that makes it accessible to readers from the AI research community—an essential quality for a position paper.

**S2**: The proposed Right to AI is a compelling perspective, particularly as AI research continues to transition into real-world applications that significantly impact society.

**W1**: The paper lacks a clear statement on the *current landscape of AI rights*. While Section 2.1 briefly mentions existing initiatives (e.g., OECD, EU), it does not provide a comprehensive overview. Given the regional disparities in AI development and regulation, the proposed framework may only be applicable to certain parts of the world. A more in-depth discussion of how AI rights vary across regions would strengthen the argument.

**W2**: The paper does not adequately address *potential limitations and unintended consequences of the Right to AI framework*. While Section 6 discusses lessons from participatory practices, I am not convinced that shifting from Proprietary Technocracy to Citizen-Controlled AI is a comprehensive solution to existing challenges. A more balanced discussion on risks and trade-offs would be valuable.

**W3**: The paper **overlooks the impact on ML researchers**, which is critical given that this position paper is aimed at the ML research community. How would the proposed rights influence AI research, model development, and the practical challenges researchers face? Addressing these concerns would make the paper more relevant to its intended audience.

**Support:**

2

---

> ### Author Rebuttal · Authors · 2025-04-01
>
> We thank the Reviewer for their constructive comments. To clarify, we envision the Right to AI not as a standalone prescription, but as an overarching framework that reinforces and precedes existing principles, rights, and legal structures. Its main goal is to raise awareness and establish inclusive norms early in AI development, supporting more specific mechanisms such as local AI councils and stakeholder audits. Below, we respond to each point and include revised paragraphs where relevant.
>
> ---
> ### Comment W1: Clearer Statement on the Current Landscape of AI Rights
>
> **Response:**
> Below is a revised paragraph inserted at the beginning of Section 2.1 to address global disparities and varied regulatory efforts:
>
> **Revised Paragraph (Section 2.1):**
> *Contemporary discourse on AI governance encompasses a broad spectrum of policy proposals, ethical guidelines, and technical approaches aimed at aligning AI systems with societal values (Mishra, 2023; Zaidan & Ibrahim, 2024; Sorensen et al., 2024). Major institutions such as the OECD and the European Union have introduced frameworks for responsible AI development, often emphasizing fairness, accountability, and transparency (Saheb & Saheb, 2024; Zhang et al., 2025). However, these initiatives vary widely in design and enforceability across different regions. For instance, while the European Union’s proposed AI Act imposes legally binding obligations for high-risk AI systems (EU AI Act, 2024), other jurisdictions frequently rely on voluntary guidelines or market-driven standards (Kerry et al., 2025). In the United States, the now-rescinded White House Blueprint for an AI Bill of Rights was primarily composed of non-binding principles, and no comprehensive replacement has emerged (The White House, 2025). In Asia, countries such as Singapore emphasize industry consultation and codes of practice, whereas Japan’s guidelines seek to harmonize technological innovation with broader societal goals (Kerry et al., 2025). Several developing countries have drafted national AI strategies or joined initiatives like the African Union’s Continental Strategy for AI, but often face structural and financial barriers that constrain robust public participation (White & Case LLP, 2024). Overall, these disparate regulatory models illustrate an ongoing tension between top-down or expert-led approaches and a growing demand for more inclusive governance. The Right to AI advanced in this paper seeks to bridge these gaps by advocating participatory frameworks, human-centered design, and collective data ownership, thus complementing rather than merely supplanting existing governance structures across diverse legal and cultural contexts.*
>
> ---
> ### Comment W2: Potential Limitations and Unintended Consequences
>
> **Response:**
> We have added a dedicated paragraph addressing Reviewer Q58X (Comments 1, 2, and 5), highlighting the limitations and practical barriers. Additional unintended consequences—such as reinforcing local power asymmetries, increased fragmentation through divergent standards, tokenistic engagement, and slower innovation due to broader deliberation—will be included in the limitations section.
>
> ---
> ### Comment W3: Impact on ML Researchers
>
> **Response:**
> We have added a paragraph in Section 7 (Recommendations) outlining how the Right to AI could affect ML researchers’ workflows and responsibilities:
>
> **Revised Paragraph (end of Section 7):**
> *As machine learning researchers are well equipped to raise awareness and morally support their surrounding communities through communication and dialogue about AI, this responsibility translates into practical steps. Implementing the Right to AI would prompt researchers to engage more systematically with non-technical stakeholders. This could involve structuring datasets with transparent documentation, designing interfaces for community feedback, and integrating diverse perspectives into model objectives. Such a shift may introduce time and resource overheads. However, it also alters the dynamics of accountability and offers opportunities to mitigate biases and strengthen public trust. Balancing technical efficiency with meaningful public engagement may require interdisciplinary collaboration, new skill sets—such as facilitation—and iterative design cycles. Although demanding, this participatory approach can enhance the relevance and robustness of AI systems while reinforcing public confidence in their development.*
>
> ---
> **These revisions respond to the Reviewer's comments by (1) clarifying the global AI governance landscape, (2) articulating potential risks and unintended consequences, and (3) highlighting practical implications for ML researchers. We believe these changes strengthen the paper’s clarity and academic rigor. Thank you**
>
> ---
> **New references**
> 1. White House (2025). President Trump on AI leadership.
> 2. Kerry et al. (2025). Network architecture for global AI policy, Brookings.
>
> Other references are in the paper.

---

> > ### Comment · Reviewer_Gwpg · 2025-04-04
> >
> > (Accidentally posted this as Official Comment before)
> >
> > I thank the authors for their rebuttal. I have read the responses, and most of my concerns have been addressed. Therefore, I will raise my score to 3.

---

### Official Review · Reviewer_Q58X · 2025-03-05

**Significance:** 4
**Argument Clarity:** 4
**Rating:** 5
**Confidence:** 4

**Questions:**

The recommendations section focuses on actionable items towards realizing the right to AI. I would be curious to understand better where the authors see areas with fundamental gaps in research to realize the right to AI at all levels of the ladder? In other words, what are the some of the big hurdles that are not primarily normative or political in nature?

**Discussion Potential:**

4

**Paper Summary:**

The paper suggests that everyone should have “a right to AI”, in particular as AI is increasingly seeping into various aspects of society. It draws parallels to “the right to the city” and leverages Arnstein’s participatory ladder to discuss the progression from minimal engagement to full communal governance of AI. It supports the positon with several examples of why the position is relevant and why it matters, before providing some actionable suggestions and presenting counterarguments in the alternate views section.

### POST REBUTTAL UPDATE

The additional remarks in the rebuttal have been helpful and I hope the authors include them in a prospective revision for the camera-ready. Overall, I retain my rating and believe the paper should be published.

**Position:**

Yes

**Position In Title:**

Yes

**Related Work:**

4

**Strengths And Weaknesses:**

**Strengths**:
* The paper is *very* well written and structured. It is intuitive to read, the paragraphs are all concise, and do not mingle with words or introduce jargon where unnecessary. It takes genuine skill to portray a rather complicated topic with several complex dimensions and still portray the gists adequately in an 8 page paper. I believe the position paper manages to do this very well.
* The position is exceptionally clear, and independent of whether one is in support of the position or not, the paper provides substantial evidence in its favor. As such, it is an excellent work that goes beyond a mere survey, but further contextualizes many interesting works in a position that’s sure to spark a lot of constructive discussion across the community. This should be the main goal of a position paper.
* The alternate views section presents actual alternate views and does not simply construct strawman style arguments, as is often the case with discussions of alternatives or limitations. This is appreciated and makes the reader genuinely reflect on what has been written before. I already found myself contemplating which parts of the position I am in full support of myself (a lot), and where I am still hesitant. At least for me, the paper has thus managed to achieve its goal already.
* Despite my only real mentioned suggestions for improvement below, which may be highly subjective in nature in the first place, I find the narrative focused and written at a level that I imagine will be interesting for people to read, even if they don’t agree with the position. Naturally the focus could be different, or there could be more details, but there is a heavy space constraint and the topical choices of the paper are highly coherent.

**Suggestions for improvement**:
* Despite section 5.4 — citizen-controlled (maximal right to AI) — suggesting to be the highest tier, it remains unclear how full communal sovereignty can be mapped to AI beyond data ownership and auditing. The latter seem to be the primary focus of examples and recommendations. What is stated supports the arguments very well, yet AI is more than the data that goes in and its resulting outcome. It’s also a lot of very complicated procedures requiring lots of highly technical expertise in algorithms, several modeling choices, learning techniques etc. It remains somewhat unclear from the paper how to strike a balance between taking diverse perspectives into account to avoid overseeing important aspects, and ensuring that sectors that require deep expertise are not unnecessarily perturbed by people “exerting their right to AI”. For instance, in education or medical, we would definitely want the right to receive universal education and healthcare, and thus have our opinions and feedback be heard, but it becomes very challenging when we “move up” the participatory ladder in such high-stakes fields. Figure 2, and the respective language used throughout the text, gives of the impression that the “lower tiers” are bad by design and generally undesirable, e.g. they are “extractive”, “proprietary”. In general, it is likely a lot more nuanced which tiers may be appropriate in which contexts, as is in parts reflected by the high context specificity in table 1 as well. This more nuanced perspective seems to be largely missing from the current position’s discussion and it thus becomes almost a bit “one note”. Some of these points are picked up in the alterantive views section, and the paper became much more well positioned at that point. However, as a suggestion, it would be nice to have some of the more nuanced takes reflected in earlier parts of the paper as well to better understand how challenging participation and a right to AI really are. This would also be important to make readers understand that any participation isn’t necessarily good participation and to prospectively avoid “participation-washing” AI systems.
* Table 1 is great and provides a lot of potential insights. Its respective discussion however, is very concise and hard to follow for readers that aren’t already aware of all the details of the surveyed projects. Perhaps a prolongued discussion could be added to the appendix.

**Support:**

3

---

> ### Author Rebuttal · Authors · 2025-04-01
>
> We thank the reviewers for their constructive feedback, which led to key revisions: we clarified the contextual limits of citizen control (Sections 2 and 5.4), expanded the discussion of Table 1 for clarity, revised Figure 2 to adopt neutral terminology, and added a concluding paragraph to Section 7 addressing technical and methodological research gaps. Detailed responses follow below.
>
> ---
>
> ### Comment 1: How deeper citizen control goes beyond data and auditing. Could there be more nuanced, context-specific approaches earlier in the text?
>
> **Response**
> Thank you for the insightful observation. To clarify our position, we have added the following paragraph:
>
> **New Paragraph for Section 5.4 (Citizen-Controlled):**
> *Citizen-controlled governance envisions significant communal authority over AI systems; however, this does not imply the exclusion of domain experts or the adoption of a uniform approach in every context. In critical fields such as healthcare and aviation, for instance, broad participation must be balanced with deep technical expertise to maintain safety and reliability. In these high-stakes domains, effective citizen control may take a hybrid form, wherein communities shape overall values and objectives while specialists guide specific technical parameters. Thus, citizen sovereignty in AI does not preclude expert collaboration; rather, it enables stakeholders to determine when and how specialized knowledge interacts with collective oversight.*
>
> ---
>
> ### Comment 2: It would be helpful to reflect this nuance earlier in the paper, so readers understand that any participation is not necessarily beneficial and that context matters deeply.
>
>
> **Response**
> We agree that participation is not inherently beneficial and that context is essential. To reflect this, we have added a paragraph at the end of Section 2.
>
> **Added Paragraph in Section 2 (Background):**
> *It is also important to recognize that participation is not universally positive and can at times become a superficial process—commonly referred to as ‘participation-washing’—if stakeholders lack meaningful influence. Moreover, certain high-stakes domains may require hybrid governance models that integrate expert oversight with varying degrees of community input. The specific form and extent of participation should therefore depend on the potential risks within a domain, the complexity of the required technical knowledge, and the capacity of communities to make well-informed decisions.*
>
> ---
>
> ### Comment 3: Table 1 is great and provides a lot of potential insights, but its respective discussion is very concise and difficult to follow if one is not familiar with all the details of the projects presented.
>
> **Response**
> We expand the discussion of Table 1 in the appendix.
>
> ---
>
> ### Comment 4: Concern that Figure 2 and related text imply lower tiers are undesirable.
>
> **Response**
> We have revised Figure 2 and its accompanying discussion to adopt more neutral terminology, including the removal of terms such as "extractive" and "proprietary" from the lower tiers.
>
> ---
>
> ### Comment 5: "What are the some of the big hurdles that are not primarily normative or political in nature?"
>
> We have added a concluding paragraph in Section 7 (*Recommendations*) to outline key research gaps that extend beyond political or normative considerations.
>
> **Added Concluding Paragraph in Section 7 (Recommendations):**
> *Beyond political and normative considerations, substantial technical gaps remain in realizing the Right to AI across all tiers. Participatory systems must offer accessible, language-inclusive interfaces that accommodate diverse forms of engagement and varying abilities, without oversimplifying complex modeling decisions. Moreover, conflict-resolution protocols require both computational and sociotechnical research to systematically integrate diverse perspectives into model design—shifting from disaggregated to aggregated viewpoints in a transparent and traceable manner. The development of explainability and interpretability tools tailored to non-experts remains in its early stages. Finally, reliable methods are needed to validate the quality and relevance of community-contributed data, particularly in regions with limited technical capacity. Addressing these challenges is essential to ensure that the Right to AI becomes not merely aspirational, but operationalized in a sustainable and equitable manner.*
>
> ---
> **We appreciate the reviewers’ acknowledgment of our manuscript’s clarity and structure, as well as their insightful feedback on how to more comprehensively address varying contexts and tiers of participation. We value the opportunity to refine our work and believe these revisions enhance the paper’s academic rigor and overall relevance.**

---

> > ### Comment · Reviewer_Q58X · 2025-04-02
> >
> > I appreciate the clarifications and additional suggested changes. I agree that these will be helpful and will further improve the paper. My rating remains strong.

---

### Official Review · Reviewer_t1DJ · 2025-03-16

**Significance:** 2
**Argument Clarity:** 2
**Rating:** 2
**Confidence:** 4

**Questions:**

1. How is the right the authors are defending different from other rights to AI that have been discussed already?

2. Can the paper's argument be made more concise and explicit?

3. What is novel or surprising about viewing AI as infrastructure? Does anyone reject this position or assume that it is false? (Section 8 seems relevant here.)

4. Can the authors delete the philosophy references mentioned above or engage more directly with their content?

5. Is the right to AI just a special case of the right to infrastructure?

**Discussion Potential:**

2

**Paper Summary:**

This paper proposes a right to AI, which “asserts that individuals and communities should meaningfully participate in the development and governance of the AI systems that shape their lives.”

This right requires that citizens not merely be consulted or placated, but that the government should partner with citizens, delegate to them, and empower them to (partly) control decisions related to AI objectives, privacy, and discrimination (pp. 2-3; Figure 1).

In practice, this calls for “a governance framework encompassing local AI councils, public audits, cooperatively managed data infrastructures, and other participatory structures that reconcile intellectual property rights with communal stewardship of AI.”

Here is how I understand the author’s main argument for a right to AI, which is inspired by Lefebvre:
1. AI constitutes social infrastructure.
2. There is a right to help develop and govern social infrastructure.
3. Therefore, there is a right to AI.

The justification for (1) is that AI meets three criteria for social infrastructure: (i) AI has broad societal impact, (ii) AI plays an essential role in daily life, and (iii) AI is a requirement for collective management (pp. 3-4).

The justification for (2) is that control over social infrastructure (e.g. public buildings and educational institutions) empowers citizens and ensures that they are not alienated from each other (p. 3).

In addition to the above argument, the authors give four justifications for the right to AI (p. 4). They distinguish four tiers of engagement in AI governance (p. 5) and explore practical implications (pp. 6-8).

## update after rebuttal

I still have the three concerns detailed in the updated portion of "strengths and weaknesses" below, namely (1) citations to philosophy, (2) the authors proposal to name the right they are interested in "the right to AI," and (3) Reviewer's 2K8h concern about novelty.

**Position:**

Yes

**Position In Title:**

No

**Related Work:**

2

**Strengths And Weaknesses:**

This paper advocates a plausible position. Here are some concerns I have:

1. The argument is not clear. If the argument is actually as simple as (1)-(3) in the summary, this can be stated explicitly, and then each premise can be defended explicitly. It is also unclear how the arguments in Section 4 relate to this main argument.

2. The authors call the right they are interested in "the right to AI," but really it is more like a right to govern AI infrastructure. There are a number of rights that might be called the right to AI: the right to explanation, the right to be treated as an individual, the right to be forgotten, and the right to contest AI (https://www.jstor.org/stable/27083420). Can the authors distinguish the right they are advocating from these other rights?

3. A few of the references are misleading. I have in mind here at least Raz 1999, Marmor 2015, Wenar 2023. These papers are cited in ways that do not really reflect their content.

4. Are the authors proposing a right to AI, or are they really proposing a right to infrastructure, and then observing that infrastructure includes AI? I have a worry about the latter. In general, we have rights that others keep the promises they make to us. In particular, we have rights that others return the books they borrowed and promised to return. If we like, we can posit a "right to received borrowed books," but note here that the right that really does the explanatory work is just the general right that others keep their promises. The "right to received borrowed books" does no explanatory work in itself. Does the right to AI have a similar role? Is it just a special case of a more general right?

I couldn't figure out how to add a "rebuttal comment," but here is a reply the authors' rebuttal:

1. I think the citation to Wenar is still inaccurate and should be removed. A power right is a right to change your other rights or other people's rights. For example, promising exercises a power right, because it gives a right to the person to whom the act is promised. The Right to AI is not a power right in this sense.

2. "The Right to AI" is a very general phrase. It commands a lot of attention as a title, but it is still unclear to me why we should call this (rather than, say, the right to contest AI decisions) the right to AI. Is the claim that the proposed right in some sense already encapsulates all of the other rights associated with AI? This would require significant argumentation. I think "a right to AI as infrastructure," "a right to AI governance," "a right to govern AI infrastructure," or something along those lines would be far more descriptive.

3. I share Reviewer's 2K8h concern about novelty.

**Support:**

2

---

> ### Author Rebuttal · Authors · 2025-04-01
>
> We thank the reviewer for their insightful and constructive comments. Below, we address each point and outline the corresponding revisions to the manuscript.
>
> ---
> ### 1. Clarification of the Arguments
> **Response:**
> We revised the *Introduction* and the beginning of Section 4 to make our main argument clearer and more explicit.
>
> **New Paragraph (Introduction):**
> *This paper’s core argument can be summarized in three steps: first, that AI increasingly functions as a form of societal infrastructure; second, that individuals and communities hold a right to help shape and govern the infrastructure that affects their lives (Benkler, 2006); and third, that therefore a “Right to AI” naturally follows. Section 3 develops the premise that AI is best conceptualized as societal infrastructure, while Section 4 provides four distinct justifications for why this right is ethically and politically warranted. In brief, we argue that: (i) to protect democratic legitimacy, impacted communities must have meaningful decision-making power over AI design; (ii) social justice demands attention to marginalized voices; (iii) epistemic autonomy requires safeguards against narrow or monolithic knowledge curation by AI; and (iv) data, which is essential for AI, is socially produced and thus merits communal oversight. In grounding these claims, our framework views AI as not merely a product of private innovation but an infrastructure of public consequence, and it aligns with global calls for more inclusive and participatory technological governance (Crawford, 2021).*
>
> ---
> ### 2. Distinguishing the Right to AI from other rights
> **Response:**
> To clarify, we add the following to the introduction.
>
> **New Paragraph (end of Introduction):**
> *We use the term “Right to AI” to emphasize the collective governance dimension of AI oversight. This governance-oriented right is broader than more familiar claims such as the right to be forgotten, the right to explanation, or the right to contest AI decisions (Kaminski, 2019; Kaminski & Urban, 2021; Zhang et al., 2024). These latter rights, while important, speak mostly to *individual* entitlements to correct or clarify AI outputs. By contrast, the Right to AI we propose extends beyond mitigating harms to actively *co-shaping* the objectives, data practices, risk thresholds, and ethical principles of AI infrastructures. In this sense, it is more accurately viewed as a “power right” (Wenar, 2023) to guide AI’s development, rather than a narrower right to information or redress.*
>
> ---
> ### 3. References Issue
> **Response:**
> - Removed *Raz (1999)*.
> - Removed *Marmor (2015)*.
> - Linked *Wenar (2023)* more explicitly to “power rights” (citing the *Stanford Encyclopedia of Philosophy*).
>
> ---
> ### 4. Right to Infrastructure vs. Right to AI
> **Response:**
> The “Right to AI,” though rooted in infrastructure governance, demands distinct focus due to its scale, opacity, and dependence on socially produced data.
>
> **New Paragraph (end of Section "Arguments for AI as Societal Infrastructure"):**
> *One might wonder if the Right to AI merely collapses into a broader right to infrastructure. While it does share foundational principles with other governance rights over public goods, AI’s particular characteristics—such as algorithmic opacity, global data reliance, and evolving automation capabilities—warrant a dedicated framework (Cohen, 2019). The Right to AI thus refines a general right to shape infrastructure, specifying the necessary mechanisms to address the unique ethical, technical, and political complexities posed by AI.*
>
> ---
> ### 5. Novelty and Objections to the Infrastructure Framing
>
> **Response:**
> We revised Sections 3 and 8 to emphasize that the framing of AI as infrastructure is contested but crucial.
>
> **New Paragraph (in Section 3):**
> *Some scholars and industry leaders dispute the premise that AI is or should be treated as public infrastructure, arguing that AI’s intangibility sets it apart from utilities like water or electricity. Others emphasize market-led solutions, suggesting that competition will naturally encourage responsible AI. However, the infrastructural viewpoint spotlights the breadth and depth of AI’s societal reach, from healthcare to education to political discourse (Plantin et al., 2018). This, we contend, justifies a governance paradigm akin to that of publicly regulated utilities. Treating AI as infrastructure thus reframes oversight as a collective undertaking, rather than a question of consumer choice or proprietary rights alone.*
>
> ---
> **We are grateful for the reviewer’s feedback, which has helped make our core argument more explicit, clarified its distinction from narrower rights, and improved our references. We hope the revisions have fully addressed the concerns raised and have improved the clarity and rigor of our manuscript.**
>
> New reference
>
> Plantin et al. (2016). Infrastructure studies meet platform studies in the age of Google and Facebook. New Media & Society

---

### Decision · Program_Chairs · 2025-04-30

**Decision:**

Accept (poster)

**Comment:**

The reviewers agree that the paper is well written and clear, and the position is very clear also. The main argument is compelling with proper evidence.